

# Risk assessment of freezing injury during overwintering of wheat in the northern boundary of the Winter Wheat Region in China

Jiahui Guo[1], Xionghui Bai[1], Weiping Shi[1], Ruijie Li[2], Xingyu Hao[1], Hongfu Wang[1], Zhiqiang Gao[1], Jie Guo[1] and Wen Lin[1]

[1] College of Agriculture, Shanxi Agricultural University, Taigu, Shanxi, China
[2] Maize Breeding Center, Zunhua Maize Breeding Institute of Hebei Province, Zunhua, Hebei, China

## ABSTRACT

Freezing injury is one of the main restriction factors for winter wheat production, especially in the northern part of the Winter Wheat Region in China. It is very important to assess the risk of winter wheat-freezing injury. However, most of the existing climate models are complex and cannot be widely used. In this study, Zunhua which is located in the northern boundary of Winter Wheat Region in China is selected as research region, based on the winter meteorological data of Zunhua from 1956 to 2016, seven freezing disaster-causing factors related to freezing injury were extracted to formulated the freezing injury index (*FII*) of wheat. Referring to the historical wheat-freezing injury in Zunhua and combining with the cold resistance identification data of the National Winter Wheat Variety Regional Test (NWWVRT), consistency between the *FII* and the actual freezing injury situation was tested. Furthermore, the occurrence law of freezing injury in Zunhua during the past 60 years was analyzed by Morlet wavelet analyze, and the risk of freezing injury in the short term was evaluated. Results showed that the *FII* can reflect the occurrence of winter wheat-freezing injury in Zunhua to a certain extent and had a significant linear correlation with the dead tiller rate of wheat ($P = 0.014$). The interannual variation of the *FII* in Zunhua also showed a significant downward trend ($R^2 = 0.7412$). There are two cycles of freezing injury in 60 years, and it showed that there's still exist a high risk in the short term. This study provides reference information for the rational use of meteorological data for winter wheat-freezing injury risk assessment.

## INTRODUCTION

Among the four major grain crops in China, only winter wheat has a growth period that spans the whole winter. Therefore, winter freezing injury is an important environmental stress that winter wheat is facing. In China, the Northern Winter Wheat Region is one of the most important wheat-producing areas, the wheat-planting area accounts for approximately 60% of the total planting area, and yield accounts for approximately 80%

Corresponding authors
Jie Guo, nxgj1115326@sxau.edu.cn
Wen Lin, slwr@sxau.edu.cn

of the total wheat yield in China (*He et al., 2011*; *Zhou et al., 2007*). Before the mid-1980s, the harsh environment of freezing temperatures and limitation of varieties caused winter wheat-freezing injury to be more serious and frequent, and the decline in population density caused by freezing injury was the main reason for wheat yield reduction (*Zhu et al., 2014*), with a decline of up to 30%–60% (*Reynolds et al., 2009*). With global warming (*Piao et al., 2010*), some varieties with strong winter habit can no longer meet the demand for low temperature to complete the vernalization process. Therefore, the northern boundary of winter wheat planting in the northern region moved northward (*Li et al., 2013*), and the variety structure of the Northern Spring Wheat Region southern boundary gradually transformed into the winter wheat. However, the frequency of extreme cold climate has increased, which also increases the probability of large-scale freezing injury at the junction area of winter and spring wheat (*Chen et al., 2017*).

Wheat-freezing injury occurs when the temperature is lower than 0 °C (*Ikkonen et al., 2020*), accompanied by various types of freezing injury, including the sudden drop of temperature in early winter, the long cold in winter, the combination of drought and freezing, and freeze–thaw in early spring (*Zheng & Qi, 1984*). When occur to freezing injury, a large number of reactive oxygen and abiotic free radicals accumulate in plants (*Soltész et al., 2011*; *Kendall & Mckersie, 1989*), that damage the cellular internal environment and membrane system, even cause the leaf wilting and tillering death (*Hassan et al., 2021*). All of these factors result in wheat not smoothly getting through winter.

Recently, freezing injury has become a hotspot in addition to drought (*Barlow et al., 2015*), waterlogging (*Herzog et al., 2016*), and saline alkali (*Kumar et al., 2015*), and many researchers have conducted in-depth studies on it (*Frederiks et al., 2012*; *Koemel Jr et al., 2004*). For example, *Zhu et al. (2018)* analyzed the relationship between multiple meteorological factors and the actual occurrence of freezing injury in Henan province, China. They obtained specific meteorological conditions for severe freezing injury and proposed that the complexity of the severe freezing injury mechanism was not explained by simple temperature data. *Zheng et al. (2015a)* and *Zheng et al. (2015b)* proposed that the absolute value of negative accumulated temperature during overwintering periods could be used as an index to divide the degree of winter wheat-freezing injury. Additionally, *Bake et al. (2006)* comprehensively considered the cold resistance ability of varieties and environmental factors, and established a wheat-freezing injury early warning system in Beijing using the decision tree method. This system described the cold resistance ability of seedlings, winter frostbite situation, and frost damage degree. *Meng et al. (2019)* comprehensively considered the influence of temperature, precipitation, and wind speed during the overwintering period, selected six freezing disaster-causing factors to establish winter wheat-freezing injury index equations for three subregions of the Northern Winter Wheat Region and conducted an overall assessment of wheat-freezing injury risk in this agroecological production zone.

Many wheat physiological models have been successfully applied in the actual production of wheat-freezing injury simulation, including the Ceres-Wheat (*Singh, Tripathy & Chopra, 2008*; *Jamieson et al., 1998*; *Liu, Yao & Jiang, 2021*), WOFOST (*Eitzinger et al., 2004*), and FROST models (*Fowler, Limin & Ritchie, 1999*; *Bergjorda, Bonesmo & Skjelvåg,*

*2008*). These models explained the mechanism of wheat-freezing injury to a certain extent, which played an important role in early warning and post-disaster evaluation of freezing injury (*Liu, Yao & Jiang, 2021*). However, the construction of these models lacks simplicity and operability. Additionally, these models are partial to theoretical calculation in the use of application as professionals should obtain data through complex software operations (*Wolf et al., 1996*). More importantly, data collection needs the help of many precision instruments, such as temperature and humidity sensors, where the accuracy of the instrument will greatly affect the accuracy of the data (*Maiorano et al., 2017*).

In this study, we comprehensively considered the climatic factors related to freezing injury, such as the water condition before overwintering, the temperature during winter, negatively accumulated temperature, and precipitation. After that, we formulated a freezing injury index (*FII*) to represent the occurrence degree of wheat-freezing injury. The applicability of the *FII* was then verified using historical data on the description of the wheat-freezing injury and the cold resistance identification data on wheat. In addition, the occurrence of freezing injury in the short term was predicted. Our results provide a theoretical basis for making full use of agro-meteorological data to assess the risk of freezing injury to guide winter wheat production.

## MATERIALS & METHODS

### Overview of the research region
Zunhua, Hebei province, China (N: 39°55″–40°22″, E: 117°34″–118°14″) was selected as the research region. It is located in the boundary of the Hebei province winter wheat-planting area, as well as by a northern boundary of the Northern Winter Wheat Region (Fig. 1). Additionally, it is the cold resistance identification test site of the NWWVRT in China. Therefore, it is of great significance to study the prevalence of winter wheat-freezing injury in this region.

### Data sources of meteorological and freezing injury information
The data of daily maximum temperature, minimum temperature, average temperature, and precipitation used in this study were obtained from 1956 to 2016 from the surface weather stations in the China Meteorological Data Network (http://data.cma.cn). Information on historical winter wheat-freezing injury was also obtained from the China Meteorological Disaster Yearbook: Hebei volume (*Zang, 2008*) and related research (*Dai et al., 2010*). The dead tiller rate data of wheat were obtained from the NWWVRT from 2007 to 2016.

### Formulation and test of the *FII*
#### Determination of the overwintering period
Winter wheat will suffer from freezing injury when the temperature drops below 0 °C. Thus, the overwintering period is determined by the standard that the daily average temperature is lower than 0 °C (*Zhang et al., 2019*). In this study, the starting and ending dates of the daily average temperature stably lower than 0 °C were calculated using a five-day moving average method–the daily average temperature for five consecutive days is lower than 0 °C
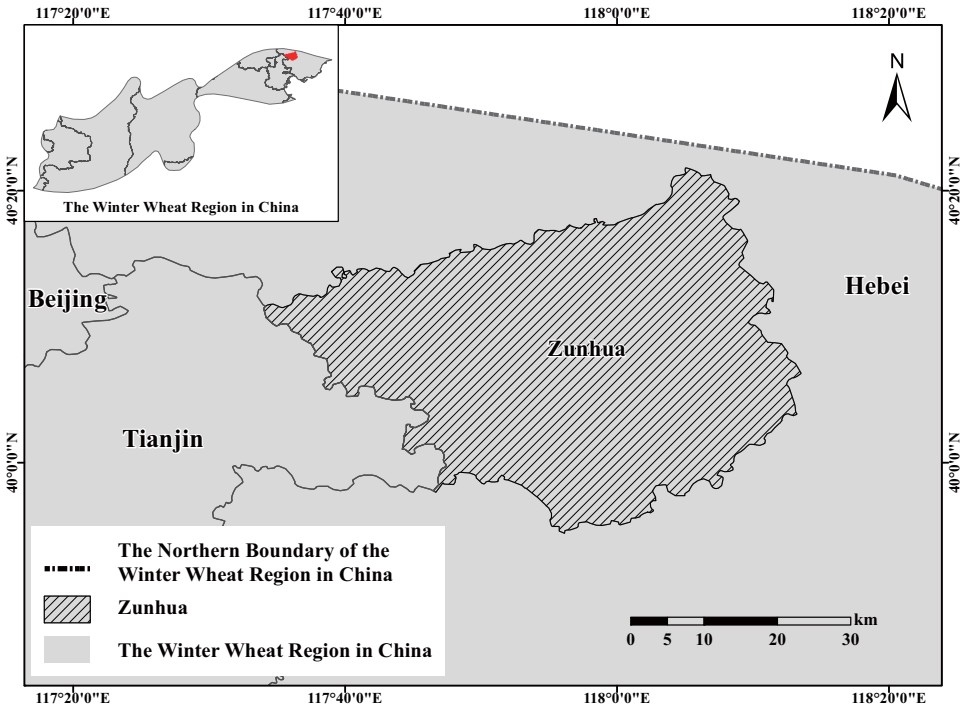

**Figure 1** Geographical coordinates of Zunhua and its location in the Northern Winter Wheat Region in China.

(*Yao et al., 2011*). Intervals define the overwintering period (*Zheng et al., 2015a*; *Zheng et al., 2015b*).

### Selection of freezing disaster-causing factors

According to meteorological data and actual climate conditions of Zunhua, seven climatic factors, including precipitation before overwintering, the extreme minimum temperature during overwintering periods, maximum-cooling range, maximum daily temperature difference, negatively accumulated temperature, average temperature, and total precipitation during overwintering periods, were selected as the freezing disaster-causing factors of winter after determining the overwintering period (Table 1). These factors include the water status before and during the overwintering period, as well as the intensity, duration, and cumulative effect of the low temperature, which can effectively reflect the mechanism of freezing injury to a certain extent.

### Formulation of the FII equation

PCA (Principal component analysis, *via* IBM SPSS v.26.0) was used to simplify the weight of the range-standardized values of the seven selected freezing disaster-causing factors (*Meng et al., 2019*). Before that, a correlation test was conducted to determine whether these data were suitable for principal component extraction (*via* IBM SPSS v.26.0). The data of the freezing disaster-causing factors were standardized after the test. Additionally, positive and negative indices are judged according to the influence of a certain index (*i.e.,*

**Table 1  Seven freezing disaster-causing factors and calculation methods.**

| Code | Disaster-causing factors | Calculation method |
|------|--------------------------|--------------------|
| $x_1$ | Precipitation before overwintering | The accumulated value of precipitation in 25 d before overwintering |
| $x_2$ | Extreme minimum temperature in the overwintering period | The minimum of extreme minimum temperature in the overwintering period |
| $x_3$ | Maximum-cooling range in the overwintering period | The maximum-cooling range of daily average temperature in continuous 72 h during cooling processes in the overwintering period |
| $x_4$ | Maximum daily temperature difference in the overwintering period | Maximum daily temperature difference in the overwintering period |
| $x_5$ | Negative accumulated temperature in the overwintering period | The cumulative value of daily mean temperature less than 0 °C in the overwintering period |
| $x_6$ | Average temperature in the overwintering period | Mean value of daily mean temperature in the overwintering period |
| $x_7$ | Total precipitation in the overwintering period | Accumulated value of daily precipitation in the overwintering period |

freezing disaster-causing factors) change on the occurrence degree of wheat-freezing injury. If the index value was larger and the influence on the freezing injury degree was greater, it showed a positive index. However, if the index value was larger and the influence on the freezing injury degree was smaller, it indicated a negative index. The positive index is calculated using formula (1), whereas the negative index is calculated using formula (2).

$$\text{Positive index}: \frac{x_0 - x_{min}}{x_{max} - x_{min}} \tag{1}$$

$$\text{Negative index}: \frac{x_{max} - x_0}{x_{max} - x_{min}}. \tag{2}$$

The positive and negative judgments of the freezing disaster-causing factor index and the choice of the range standardization formula are shown in Table S1.

Furthermore, the range-standardized values of seven disaster-causing factors were analyzed using PCA, and principal components whose eigenvalues > 1.0 were extracted. The weight coefficient of each freezing disaster-causing factor was then calculated using the sum of the proportion of the corresponding eigenvalues of each principal component and its variance product to the total variance of the extracted principal components $a_i(a_1-a_7)$. Finally, the weighted summation of formula (3) is substituted to obtain the principal component synthesis expression, *FII*. The calculation formula of *FII* is as follows:

$$FII = \sum_{i=1}^{s} a_i x_i \tag{3}$$

where *FII* is the annual freezing injury index, $x_i$ is the range-standardized value of a freezing disaster-causing factor, and $a_i$ is the weight coefficient of the corresponding freezing disaster-causing factor.

## Fit-test of *FII* and freezing injury degree

Existed studies (*Fuller et al., 2007*) mostly have used the dead plant rate (the ratio of the dead plant to the total plant) to determine the degree of wheat-freezing injury. However, this situation is limited to severe freezing injury. When freezing injury is mild, whole plant death rarely occurs, and it is the death of some tillers of the plant (*Limin & Fowler, 2006*). Therefore, the ratio of dead tillers to total tillers was used as the index to judge the degree of freezing injury in this study. To explore the relationship between the *FII* and the actual freezing injury degree, *FII* and data on the tiller death rate of Zunhua NWWVRT from 2007 to 2016 were used for univariate linear regression analysis (*via* IBM SPSS v.26.0).

## Wavelet analysis

Wavelet analysis can indicate the periodic characteristics of time series information in different timescales, and it is better to show the change trend and qualitatively predict the future development trend (*Kumar & Foufoula-Georgiou, 1993*). In this study, the Morlet wavelet (*Peng et al., 2013*) was used to analyze the *FII* data from 1956 to 2016 (*via* MATLAB 2020b). As the time series data in this study were limited time series, the symmetry extension method was used to extend the two ends of the series to eliminate or reduce the possible "edge effect". The Morlet complex wavelet function in the wavelet toolbox of MATLAB was then used to transform the extended data sequence, calculate the real part of the wavelet coefficients, and draw the real part diagram of the wavelet coefficients. The wavelet toolbox was further used to calculate the wavelet variance and to draw the wavelet variance and main period wavelet coefficient diagrams.

# RESULTS

## The starting and ending dates and the change trend of the overwintering period in each year

The starting and ending dates of the daily average temperature stably lower than 0 °C (*i.e.,* the starting and ending dates of the overwintering period in each year) in each year were obtained from the five-day moving average method (Fig. 2). According to the change in the starting and ending dates of the overwintering period over the years, the starting dates were gradually delayed (Fig. 2A), and the fluctuation range was from the middle and late November to early December and was gradually smaller. These results indicate that the start date of the overwintering period was gradually stable in early December. Additionally, the end date was gradually advanced (Fig. 2B), and the fluctuation range was advanced from the middle and early March to the middle and late February, but the fluctuation range was large, indicating that the end date of overwintering was unstable.

## Formulation and verification of the *FII* equation
### Correlation between freezing disaster-causing factors

Information on the seven freezing disaster-causing factors in the overwintering period of each year was calculated, and a correlation test was conducted (Fig. 3). Results show that there was a certain correlation among disastrous factors, except for the "precipitation before overwintering". These results indicated that a comprehensive variable can be

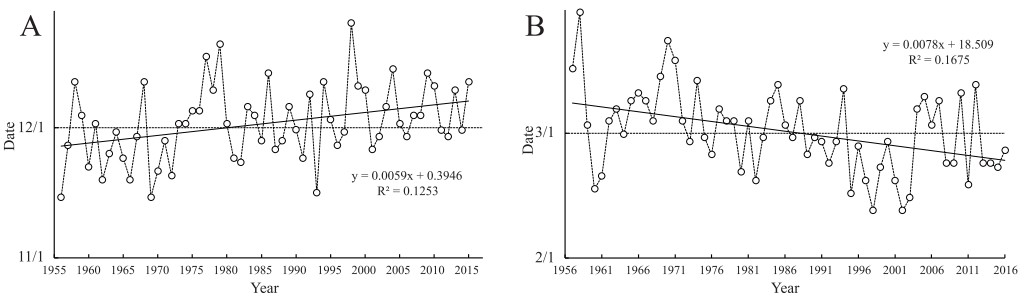

**Figure 2   The starting and ending dates of the overwintering period.** (A) The starting date of the overwintering period. (B) The ending date of the overwintering period.

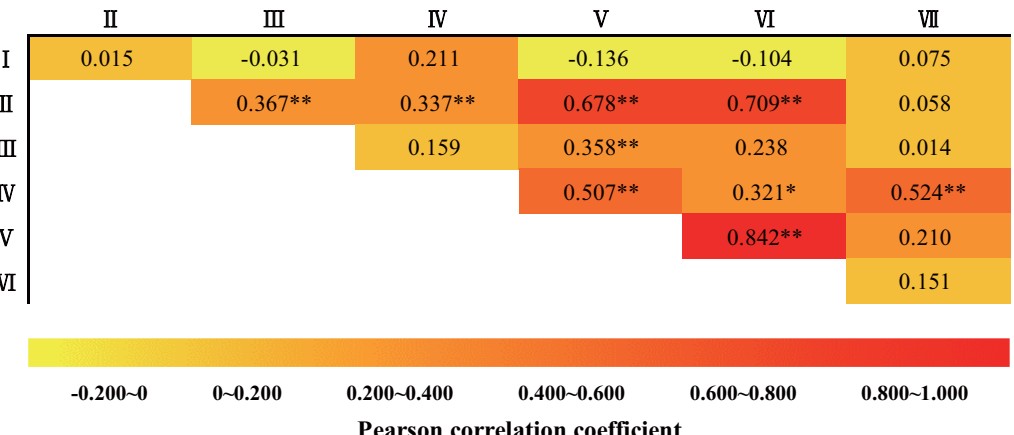

**Figure 3   Correlation test of freezing disaster-causing factors.** I: Precipitation before overwintering; II: extreme minimum temperature during overwintering periods; III: maximum-cooling range during overwintering periods; IV: maximum daily temperature differences during overwintering periods; V: negatively accumulated temperature during overwintering periods; VI: average temperature during overwintering periods; VII: total precipitation during overwintering periods. An asterisk (*) indicates a significant correlation at the 0.05 level, and two asterisks (**) indicate a significant correlation at the 0.01 level.

constructed by the PCA method to reflect the information contained in the seven freezing disaster-causing factors.

### Formulation of the FII equation

KMO and Bartlett sphericity test showed that the data were suitable for PCA (KMO = 0.639, $P = 1.18E–25$). In total, two effective principal components were extracted, and the total variance interpretation rate was 62.85%. The weight coefficient *FII* in formula (3) was calculated from the sum of the proportion of the product of the eigenvalue corresponding to the principal component and its variance to the total variance of the extracted principal component.

Eventually, it can be seen from formula Eq. (4) that when the more range-standardized value of freezing disaster-causing factors, the more *FII* value, indicated that the more risk

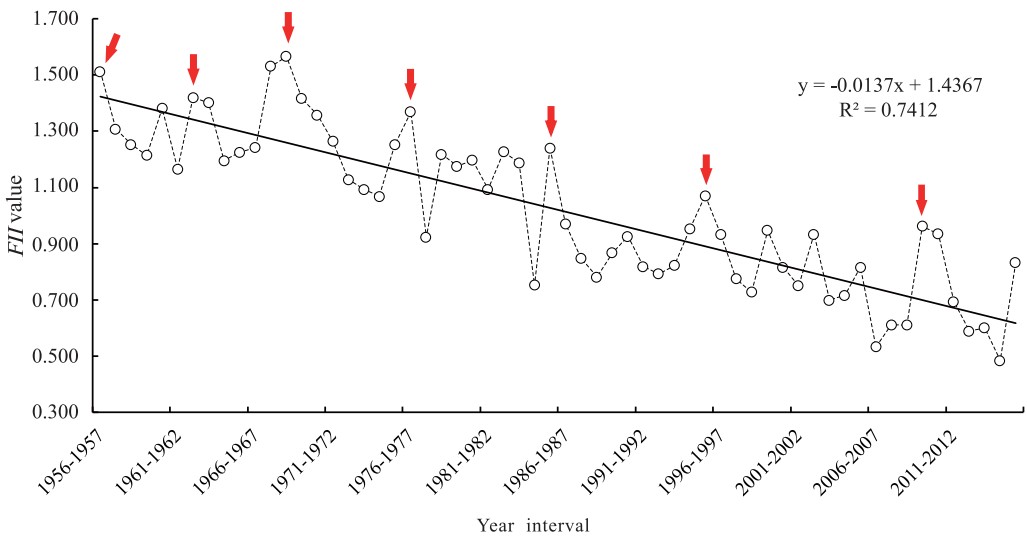

**Figure 4   FII value and changes in Zunhua from 1956 to 2016.** Red arrow represents the year in which the frost damage occurred in the historical record.

probability of severe freezing injury.

$$FII = 0.144x_1 + 0.262x_2 + 0.119x_3 + 0.413x_4 + 0.326x_5 + 0.278x_6 + 0.331x_7. \qquad (4)$$

## Verification of *FII* values and wheat-freezing injury

The range-standardized values for seven freezing disaster-causing factors in each year were substituted into formula Eq. (4) to obtain *FII* values for each year (Fig. 4). Combined with the occurrence of wheat-freezing injury in history (*Zang, 2008*), our results showed that *FII* values were higher during the severe freezing injury year (the year indicated by the arrow in Fig. 4). These values better reflect the situation of historical freezing injury, which indicated that the *FII* can be used to reflect the actual situation of winter wheat-freezing injury.

On the basis of the cold resistance identification of the Zunhua NWWVRT, univariate linear regression was used to analyze the relationship between the dead tiller rate and *FII* values. Regression analysis showed that the dead tiller rate and *FII* value passed the 0.05 significance test ($P = 0.014$), indicating that a reliable functional relationship can be established between the two indicators. According to the histogram of the residual error of the regression analysis model used, the overall distribution of the residual error was normal. Combined with a P–P diagram, the regression model has a good fit (Fig. 5). The linear regression equation of univariate was calculated as follows:

$$y = -34.826 + 65.511x, \qquad (5)$$

where $x$ and $y$ represent the *FII* index and dead tiller rate, respectively.

Regression analysis results also showed that the larger the *FII* value was, the higher the dead tiller rate was and the more severe the degree of freezing injury was. It showed
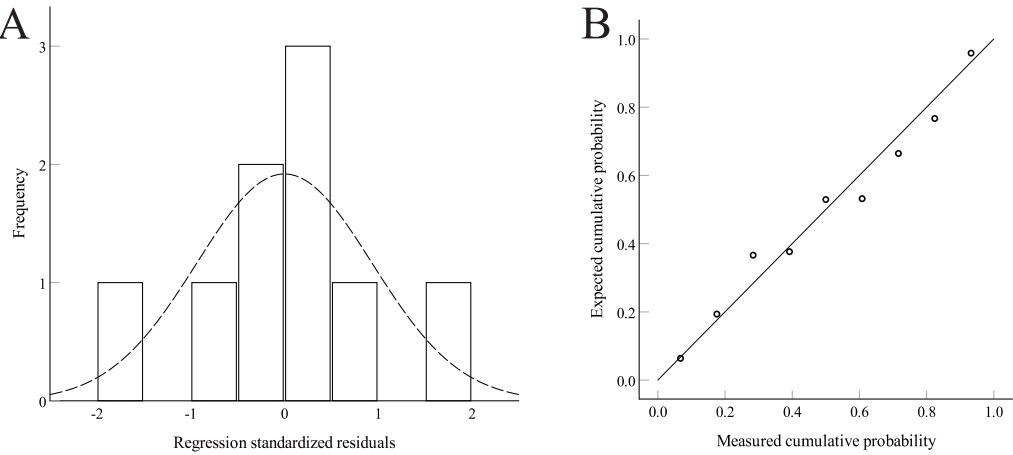

**Figure 5** **The residual histogram and P–P plot of the unary linear regression model.** (A) Residual histogram. (B) P–P plot.

that the *FII* value was a good representation of the degree of freezing injury and had high applicability.

## Analysis on the change trend and causes of the *FII* values

As the *FII* values can reflect the degree of freezing injury, it is of practical significance to analyze the change trend of *FII* values in 60 years. As the result shows, the interannual variation range of the *FII* values in Zunhua is 0.485–1.566 from 1956 to 2016, with the maximum value from 1968 to 1969 and the minimum value from 2014 to 2015 (Fig. 4). From the results, the amplitude of the *FII* between 1956 and 1986 was larger than 1987 to 2016, and there are many peaks during this period, which indicated that wheat-freezing injury occurred frequently in this period. By contrast, from 1987 to 2016, the amplitude of the *FII* was small, and the frequency of freezing injury was less. Observed the change trend of the *FII* curve during this period, 1995–1996 and 2009–2011 had the two largest peaks where severe freezing injury occurs. From overall view, the interannual variation trend of the *FII* in Zunhua showed a significant downward trend ($R^2 = 0.7412$) in the 60-year period from 1956 to 2016, but the interannual fluctuation range within whole period was large, with a coefficient of variation value of 27.2%. At the end of the curve, the fluctuation range of the *FII* increased, which indicates that the risk probability of freezing injury has also increased.

For the change of the overwintering period in 60 years, the delay in the starting time and advance in the ending time of the overwintering period (Fig. 2) led to the shortening of the overwintering period (Fig. 6A). The extreme minimum temperature (Fig. 6B), average temperature (Fig. 6C), and negatively accumulated temperature (Fig. 6D) during the overwintering period also showed an increasing trend. This result show that the gradual warming of the overwintering environment of winter wheat, which is conducive to safe overwintering. Correspondingly, the change in the *FII* over the years showed a decreasing trend (Fig. 4). Although the climate was gradually warming in the general environment,

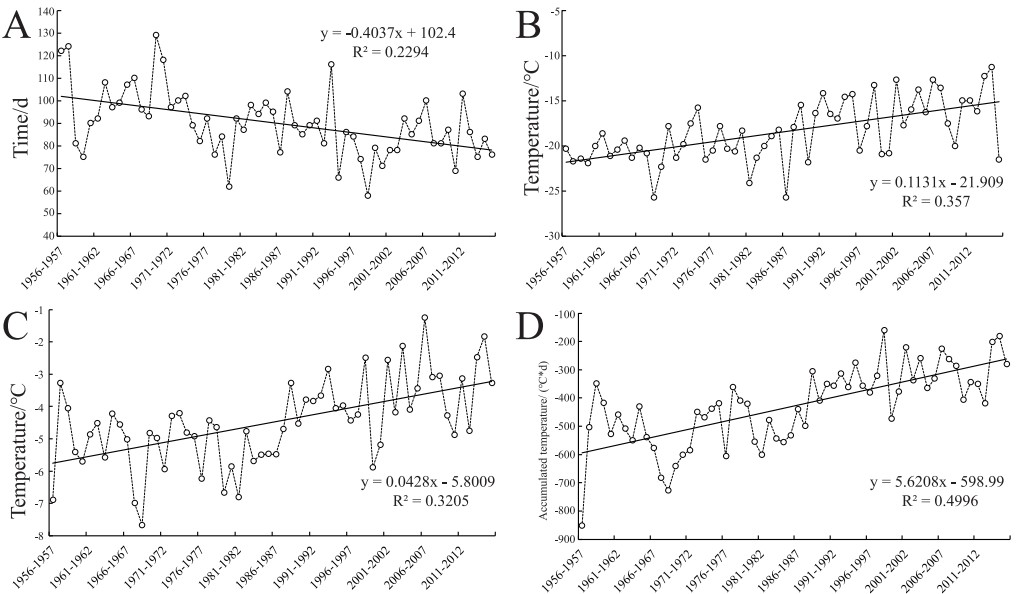

**Figure 6  Changes in climate factors during the wintering period for the past 60 years.** (A) Duration of the overwintering period; (B) extreme minimum temperature during overwintering periods; (C) average temperature during overwintering periods; (D) negatively accumulated temperature during overwintering periods.

the frequency of extreme climate increased recently, and the risk of wheat-freezing injury increased.

## Wavelet analysis

Three periods, including short (0–10a) ("a" means the timescale "year"), medium (11–20a), and long periods (21–35a), were identified in the evolution process of the *FII* (Fig. 7A). The long period (21–35a) spanned the whole timescale, showing obvious seasonal changes in abundant–withered–abundant, with peaks in 1963, 1985, and 2007, respectively. The periodicity of the other two periods was not particularly significant, but the characteristic of the short period (0–10a) was more obvious than that of the medium period (11–20a). In the wavelet variance curve (Fig. 7B), the fluctuation energy of the *FII* time series also has a clear peak on the timescale of 33a, which indicates that the periodic oscillation around 33a was the strongest. This was the first main cycle of the variation. Similarly, a low peak value on the timescale of 4a was also observed, which is the second main cycle of annual variation. Additionally, the average period of change was approximately 40 years on the 33a timescale, and it has experienced approximately 1.5 abundant–withered periods (the red curve in Fig. 7C). On the 4a timescale, the average period of change was approximately five years, which has experienced approximately 12 abundant–withered periods (the blue curve in Fig. 7C).

As a whole, the real part of the wavelet was not completely closed at the right end (Fig. 7C), which indicates that it was still in the period of high risk of freezing injury. Additionally, the right end value of the first principal period curve of the periodic wavelet

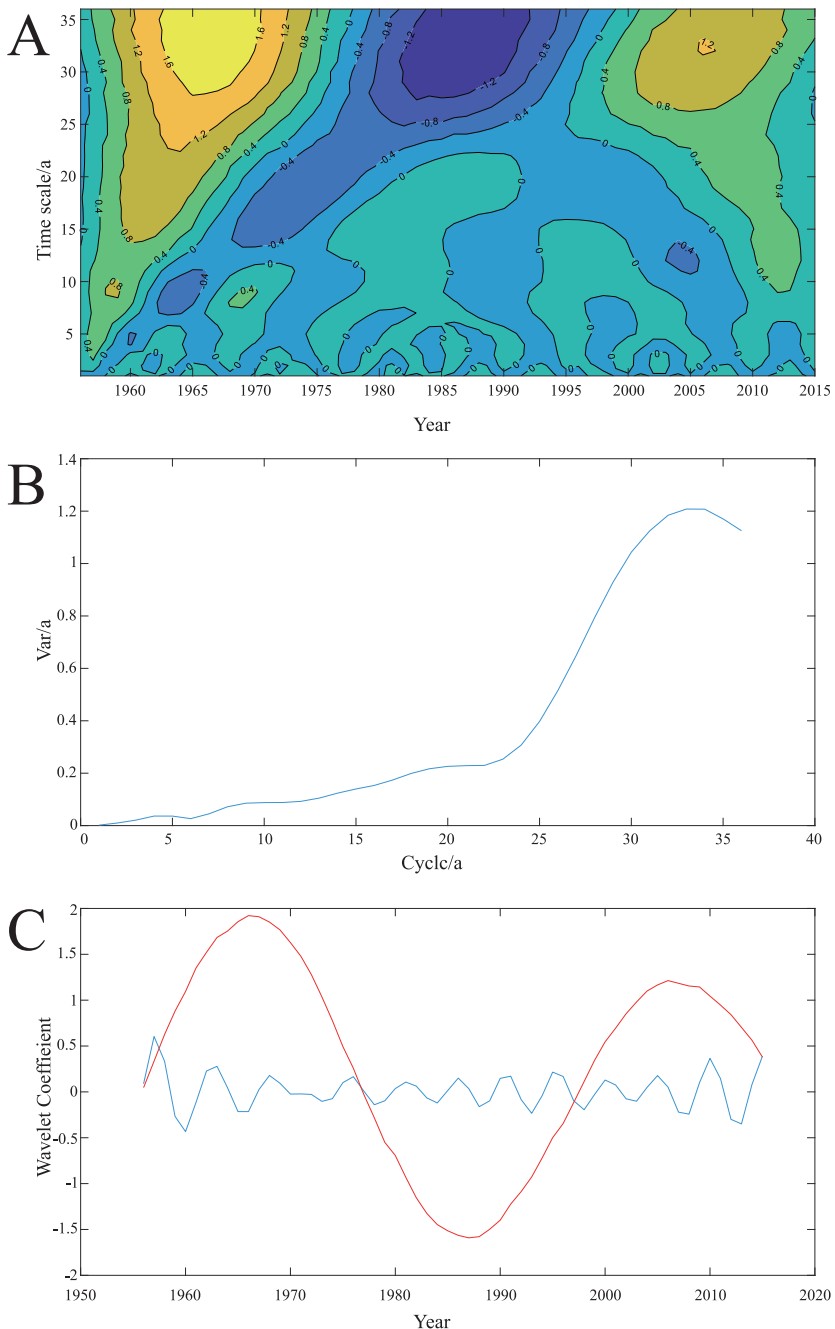

**Figure 7** **Results of *FI* value changes in 60 years by Morlet wavelet analysis.** Morlet wavelet is used to transform the *FII* value from 1956 to 2016, and the result of the wavelet analysis was drawn. (A) Wavelet coefficient real part graph. Abscissa, and ordinate represents the year and timescale (a), respectively. When the isoline is positive, it indicates that the degree of frost damage was serious. When the isoline is negative, it indicates that the degree of frost damage was light. The zero line indicates the turning point of the degree of frost damage. (B) Wavelet variance graph. The peak value indicates that the periodic oscillation is strong and the periodic characteristic is remarkable. (C) Periodic wavelet coefficient graph. The red curve represents the 33a timescale, and the blue curve represents the 4a timescale.

coefficient graph did not drop below zero, and the second principal period was at its peak, which indicates that the fluctuation energy of the freezing injury probability was strong in a short term. This result also indicates that the risk of wheat-freezing injury probability was high.

## DISCUSSION

### Occurrence and evolution of freezing injury and its climatic characteristics

Because the *FII* is a better representation of the real freezing injury situation (Figs. 4 and 5), the occurrence and evolution law of winter wheat-freezing injury in Zunhua in the past 60 years can be analyzed according to the change trend in *FII*. From the change in *FII* within 60 years, the freezing injury of winter wheat in the first 30 years showed the characteristics of heavy degree, high frequency, and large interannual fluctuation, indicating that the climate in winter before the mid-1980s was harsh and that climate change was unstable (*Zhu et al., 2014*). The low temperature and absolute value of the negatively accumulated temperature during the overwinter period were also large, and the frequency of extreme low temperature was high. These conditions were not conducive to winter wheat, which further led to a decrease in population density and yield reduction (*Livingston & Swinbank, 1962*; *Fowler et al., 1996*).

After the mid-1980s, the occurrence degree and frequency of freezing injury decreased (*Tao et al., 2006*), which was proposed to be caused by global warming (*Zhai et al., 1999*). In this study, the delay of the starting date and the advance of the ending date of the winter wheat overwinter period in Zunhua for the past 60 years led to the shortening of overwinter time. The negatively accumulated temperature, average temperature, and extreme minimum temperature of the overwinter period also showed an increasing trend (Fig. 6), which further confirmed the inference that climate warming reduced the risk of wheat-freezing injury.

Since the beginning of the 21st century, climate instability has increased (*Tian et al., 2016*). Additionally, extreme climate events occurred frequently (*Hänninen, 1991*), and the probability of winter wheat-freezing injury increased (Fig. 4). Our results found that the occurrence of freezing injury in Zunhua over the past 60 years showed a periodic change, and the small period was nested in the large period. At present, *FII* curve is at the end of a long period-positive peak and the top of the short period-positive peak (Fig. 7C), indicating that the probability of wheat frost damage in this area in the future will be relatively high. Therefore, to avoid severe freezing injury of winter wheat, it is necessary to select wheat varieties with strong cold resistance in future production processes, which can greatly reduce the impact of periodic freezing injury on grain production.

### Guidance of risk assessment of freezing injury to agricultural production

Our results showed that the probability of freezing injury in Zunhua was high in the short term (Fig. 7C). On the basis of these results, the following problems in the production process should be paid attention to: (1) the climate in the edge area of the original planting

strong winter wheat cannot meet the needs for wheat complete the vernalization process because of the gradual warming of the climate. Previous studies have shown that the wheat variety structure in northern boundary of Winter Wheat Region in China was gradually changing from a strong winter to winter or semi-winter (*Zheng, Ge & Hao, 2002*; *Wang et al., 2013*), which will increase the risk probability of seedlings having freezing injury when they encounter extreme low temperature weather (*Paulsen & Heyne, 1962*). Therefore, growers should try selecting varieties with stronger winter characters to reduce the adverse effects caused by climate change. (2) Owing to the delay in the starting date of the wheat-overwintering period (Fig. 2A), the sowing time should be appropriately late according to the actual climate conditions to prevent the vigorous growth of seedlings before overwintering (*Sadras & Monzon, 2006*). Simultaneously, irrigation should be timely and sufficient before winter to avoid freezing injury caused by drought (*Fischer, 1985*). When breeding new winter wheat varieties, cold resistance should also be screened (*Zheng et al., 2015a*; *Zheng et al., 2015b*).

## Suggestion on identification of cold resistance of varieties

Owing to global warming, cold resistance identification of new varieties in a field will result in a situation where the dead plant and dead tiller rate of all varieties are zero (*Wang et al., 2008*). Here, it was impossible to make an objective evaluation on the cold resistance of all varieties. Therefore, to fully reflect on the cold resistance of various varieties, the occurrence probability of freezing injury can be improved by the following ways: (1) sowing all tested varieties in advance to form exuberant seedlings before overwintering, which can increase the probability of freezing injury of seedlings, or decrease planting density, which will increase the probability of freezing injury (*Fowler & Limin, 2004*). (2) Due to the north boundary of Winter Wheat Region is constantly moving north, the area with higher latitudes and worse wintering environment can also be added as a new cold resistance identification test point to fully reflect the cold resistance ability of various varieties(*Chen et al., 2011*; *Yang, Liu & Chen, 2011*).

## Deficiency and suggestion of establishing the *FII* equation

The occurrence of winter wheat-freezing injury is an extremely complex process (*Craufurd & Wheeler, 2009*). The seven freezing disaster-causing factors selected in this study for formulating a *FII* equation cover effective information such as the water condition before overwintering, the intensity, duration, and cumulative effect of low temperature, as well as the effective precipitation during overwintering. The *FII* can reflect the occurrence degree of winter wheat-freezing injury through the test. Our results showed that, the higher the *FII* value is, the more severe the freezing injury would be. In addition to the selected factors causing freezing injury, the occurrence of freezing injury is affected by wheat genetic factors (*Yan et al., 2004*; *Fu et al., 2005*), duration and thickness of snow cover (*Yoshida et al., 1998*), duration and speed of wind (*Porter & Gawith, 1999*), and cultivation management methods (*Hassan et al., 2021*). For example, during the overwintering period of 2012–2013, wheat seedlings were covered with snow for approximately one month, which effectively protects seedlings; hence, the freezing injury degree of seedlings was less,

but the *FII* value was higher. Thus, more meteorological factors related to wheat-freezing injury can be included in the formulation of a *FII*. However, it should also be noted that the increase in data collection make this model more complex, which is not conducive to the operation of users.

Additionally, soil moisture condition has a great impact on wheat overwintering as drought aggravates the occurrence of freezing injury (*Al-Issawi et al., 2013*). The content of soil moisture during overwintering directly affects the occurrence probability of wheat-freezing injury (*Alireza et al., 2021*). We speculate that taking soil moisture content as the water condition factor of *FII* equation probably achieve more accurate prediction effect. In conclusion, the establishment of the *FII* equation needs to be further improved.

## CONCLUSIONS

Seven freezing disaster-causing factors were chosen to formulated *FII* in Zunhua from 1956 to 2016. The results show that *FII* can reflect the freezing injury situation in Zunhua over the years. According to the analysis of the climatic factors during the overwintering period, the delay in the start date and the advance of the end date led to shortening of the overwintering period. Additionally, the climate conditions were gradually mild, and the occurrence degree of winter wheat-freezing injury showed a significant downward trend ($R^2 = 0.7412$). Wavelet analysis shows that the change of *FII* has clear periodicity in the whole timescale. And it can be predicted that Zunhua is still in a high-risk period of freezing injury.

## ACKNOWLEDGEMENTS

We gratefully acknowledge help from Prof. Robert A McIntosh, University of Sydney, with English editing.

### Funding
This work was supported by grants from the National Key R&D Program of China (2017YFD0101000), the National Natural Science Foundation of China (31901541), and the Natural Science Foundation of Shanxi Province (201901D211361). The funders had no role in study design, data collection and analysis, decision to publish, or preparation of the manuscript.

### Grant Disclosures
The following grant information was disclosed by the authors:
National Key R&D Program of China: 2017YFD0101000.
National Natural Science Foundation of China: 31901541.
Natural Science Foundation of Shanxi Province: 201901D211361.

### Competing Interests
The authors declare there are no competing interests.

## Author Contributions

- Jiahui Guo and Wen Lin conceived and designed the experiments, analyzed the data, prepared figures and/or tables, authored or reviewed drafts of the paper, and approved the final draft.
- Xionghui Bai analyzed the data, prepared figures and/or tables, authored or reviewed drafts of the paper, and approved the final draft.
- Weiping Shi, Ruijie Li, Xingyu Hao, Hongfu Wang and Zhiqiang Gao analyzed the data, authored or reviewed drafts of the paper, and approved the final draft.
- Jie Guo conceived and designed the experiments, analyzed the data, authored or reviewed drafts of the paper, and approved the final draft.

## Data Availability

The raw data is available in the Supplementary File.

## Supplemental Information

Supplemental information for this article can be found online at http://dx.doi.org/10.7717/peerj.12154#supplemental-information.

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
