# Peer review of "Risk assessment of freezing injury during overwintering of wheat in the northern boundary of the Winter Wheat Region in China"

_PeerJ, doi:10.7717/peerj.12154_

## Round 0.1 · original submission · Minor Revisions

The manuscript has been reviewed by two experts. Both of them think it is interesting research and suitable for our journal. I suggest a minor revision based on their review comments. Please modify your manuscript accordingly.

Reviewer 1 ·

Basic reporting

This manuscript is well written; however, some language mistakes still exist, it should be checked throughout the whole text. The manuscript structure is clearly arranged and meets the requirements of the Journal. And the figures and tables are related to the content. The manuscript can be accepted after some minor revisions.

Experimental design

The research on the climatic conditions of wheat freezing injury in the past is helpful to assess the risk of winter wheat freezing injury and increase grain yield. The research in this manuscript is based on 60 years of meteorological data in Zunhua and historical records of wheat freezing injury. A freezing injury prediction model was established and validated. The research question is well defined, which is within the aims and scope of the journal.

Validity of the findings

The research proved that this model is suitable for winter wheat-freezing injury prediction in different years, and provided a new reference for risk assessment of wheat freezing injury. The results are well presented and well supported by the measurement.

Additional comments

Abstract
1. Line 28: The abbreviation of "freezing inventory index" should be "FII" instead of "FI". After the abbreviation appears for the first time, the full name does not need to appear in subsequent places. Please make a unified expression.
2.Line 33: The expression "in the future" is not accurate and should be "in the short term".
3.Line 39: The expression of "freezing injury" or "frost damage" in the full manuscript should be unified.
Main part
1. The introduction needs revisions. The newest reference is from 2019, making the manuscript not up to date with the current literature. I strongly suggest a thoroughly review and update of the literature. Not only for the introduction but also for the discussion.
2. Line 56: The expression of "the variety structure of the northern spring wheat-planting southern area gradually transformed into the winter wheat" is unclear, and the first letter of wheat region name should be capitalized.
3. Line 82: The expression "in this study" is incorrect. Here is the information in the quoted literature, not in this research.
4. Line 91: The word "postdisaster" should be "post-disaster".
5. Line 107: In the title of this section, "research site" should be changed to " research region"
6. Line 119: Using "tiller" instead of "stem" can obtain better expression accuracy.
7. Line 127: The expression "passing through 0 ℃" is inappropriate and should be "stably lower than 0 °C".
8. Line 135: The expression of "freezing disaster-causing factors" should be unified in the full manuscript.
9. Line 168: The expression of "the death rate at the seedling stage" is incorrect and should be "the dead plant rate".
10. Line 181: "Morlet" is a proper noun and should be preceded by "the".
11. Line 190: When referring to the software used, it should not be divided into separate sections, but should be displayed in combination with the method.
12. Line 198: Note that the use of words should be consistent before and after the full manuscript, such as "starting and ending dates" or "starting and ending times".
13. Line 241: If the two variables pass the correlation significance test in univariate linear regression, it indicates that there is a credible linear relationship between the two variables, rather than significant correlation.
14. Line 277: The "time scale" should be replaced by "timescale".
15. Line 385: "Conclusions" instead of "conclusion".
16. Line 386: The conclusions are a concise part of the full manuscript, which needs to be further modified.
17. The authors should carefully examine the grammatical errors in the manuscript.
18. In Figure 7C of wavelet analysis, what do the red and blue curves represent respectively? Therefore, for each picture, valid information should be displayed in the annotation.
19. In the title of Table 1, "disaster causing factors of freezing damage" should be replaced by "freezing disaster-causing factors".

·

Basic reporting

Currently, a brief introduction to the research is sufficient. The content of the manuscript is clearly arranged and meets the requirements of peer J. However, there are still some problems that should be corrected; hence a minor revision is suggested for this manuscript.

Experimental design

Wheat freezing injury is often a restriction for winter wheat production. Understanding how climatic conditions impact wheat freezing injury in the past is of great importance for assessing the risk of winter wheat freezing injury. The freezing injury prediction model established in this research is useful for avoiding freezing injury in the research area.

Validity of the findings

This research established a freezing injury prediction model and proved that this model is suitable for injury prediction in different years. I think the result is valid.

Additional comments

1. The literature review and discussion in this MS should keep pace with the latest literature published, while currently the latest literature was published in 2019.
2. "freezing inventory index" should be "FII" rather than "FI". Please revise it in the whole MS.
3. Line 71: The content description of the cited literature should be concise.
4. Line 114: The title of this section should be changed to "Data sources of meteorological and freezing investigation information" to make the information more detailed.
5. Line 157: The expression "were greater than one" is inappropriate and should be expressed directly with "> 1.0".
6. Line 204: Reference should be made when describing the pictures in the group diagram separately, such as (Fig. 2A).
7. Line 246: The expression of "FII index" is inappropriate, "FII value" should be used instead.
8. Line 270: "a" should be replaced by "an".
9. Line 324: The description of " under the premise of ensuring yield, quality, and disease resistance " is unnecessary.
10. Line 333: "It was observed from the results that" is redundant with the description in the previous sentence and should be removed.

---

## Round 0.2 · accepted · Accept

Thank you for your revision and I am happy with your responses to review comments. Congratulations and welcome to submit your manuscript to our journal again in the future!